evolution, ecology, cognition

cognition, ecology, evolution, learning, primates, habitat

**Author for correspondence:**
Johanna Henke-von der Malsburg
e-mail: jmalsburg@dpz.eu

# Linking cognition to ecology in wild sympatric mouse lemur species

Johanna Henke-von der Malsburg[1,2,3], Peter M. Kappeler[1,3] and Claudia Fichtel[1,2]

[1]Behavioral Ecology and Sociobiology Unit, German Primate Center, Leibniz Institute for Primatology, Göttingen, Germany
[2]Leibniz ScienceCampus 'Primate Cognition', Göttingen, Germany
[3]Department of Sociobiology/Anthropology, Johann-Friedrich-Blumenbach Institute of Zoology and Anthropology, Georg-August-University Göttingen, Göttingen, Germany

  JH-vdM, 0000-0001-6055-8506; PMK, 0000-0002-4801-487X; CF, 0000-0002-8346-2168

Cognitive abilities covary with both social and ecological factors across animal taxa. Ecological generalists have been attributed with enhanced cognitive abilities, but which specific ecological factors may have shaped the evolution of which specific cognitive abilities remains poorly known. To explore these links, we applied a cognitive test battery (two personality, ten cognitive tests; *n* = 1104 tests) to wild individuals of two sympatric mouse lemur species (*n* = 120 *Microcebus murinus*, *n* = 34 *M. berthae*) varying in ecological adaptations but sharing key features of their social systems. The habitat and dietary generalist grey mouse lemurs were more innovative and exhibited better spatial learning abilities; a cognitive advantage in responding adaptively to dynamic environmental conditions. The more specialized Madame Berthe's mouse lemurs were faster in learning associative reward contingencies, providing relative advantages in stable environmental conditions. Hence, our study revealed key cognitive correlates of ecological adaptations and indicates potential cognitive constraints of specialists that may help explain why they face a greater extinction risk in the context of current environmental changes.

## 1. Background

The evolution of cognitive abilities has been linked to variation in brain size, which covaries across species with social factors (social intelligence hypothesis [1]) and/or ecological challenges (ecological intelligence hypothesis [2]). Recent comparative analyses across primates suggested that evolutionary variation in brain size is better predicted by ecological than social factors [3]. Yet, little is known about whether and how these factors are linked to performance in cognitive tests in primates, but also across other taxonomic groups [4,5]. Hence, to better understand the evolution of cognitive abilities and the underlying variation in brain size, studies of how variation in specific ecological or social factors are linked to performance in cognitive tests across taxa are required.

In this context, the degree of ecological specialization has been suggested to covary with cognitive abilities (opportunistic intelligence hypothesis [6]). This notion builds upon the idea of characterizing a species's ecological niche as a multidimensional space that combines all adaptations to ecological conditions that contribute to its evolutionary success [7]. Accordingly, an ecological generalist experiences a wider niche breadth than a specialist [8].

Generalist species are assumed to be better and more flexible learners than specialists [9,10]. Since generalists should be exposed to a greater variety of ecological conditions, they may also face a greater variety of ecological problems. Hence, they may have evolved specific innovative problem-solving abilities to overcome various problems. Similarly, the diverse ecological conditions may create a need for greater behavioural flexibility, especially when conditions change unexpectedly

[11]. Hence, species that experience harsher or more dynamic environmental conditions are more flexible or more innovative than others [12]. Innovative abilities and flexibility therefore appear to be tightly linked [13]. Furthermore, innovation appears to be positively correlated with other cognitive abilities [14]. In addition, more innovative or behaviourally flexible species experience greater colonization success [15] or a greater diversification potential [16]. Both evolutionary processes have been linked to the evolution of larger brains, especially when colonizing seasonal regions [17]. Finally, dietary generalists have indeed larger brains than dietary specialists [18–20]. Despite these suggestive links, generalists, do not consistently perform better in cognitive tests than specialists, however [4].

To systematically examine covariation between cognitive abilities and the degree of ecological specialization, we applied a comprehensive cognitive test battery to wild individuals of two mouse lemur species (*Microcebus* spp.) that vary in some of their ecological adaptations but share key features of their social systems. Grey (*M. murinus*, GML) and Madame Berthe's mouse lemurs (*M. berthae*, MBML) represent separate lineages within the genus *Microcebus* that shared a common ancestor as early as 9–10 Ma [21]. The comparison of these two species is informative because they are both nocturnal solitary foragers that are syntopic, and therefore experience identical current environmental conditions, but MBML is ecologically more specialized [22,23]. Such direct comparisons of cognitive performance in pairs of sympatric sister species can help to reveal the role of ecological factors in the evolution of cognition.

GML inhabit various habitat types across western Madagascar, occur in primary as well as secondary forests, and even in highly degraded forest fragments [24,25], making them habitat generalists. Their feeding niche breadth, based on Levin's standardized index, has been estimated as 0.63 [23], supporting this classification. MBML occur only in a few square kilometres of seasonally dry deciduous lowland forests [26] and have an annual feeding niche breadth of 0.12 [23], qualifying them as habitat specialists. As the smallest living primates, they are also more sensitive to natural and anthropogenic habitat modifications [26], markedly decreasing their population size in recent years [27], and justifying their classification as 'Critically Endangered' [28].

Using a comprehensive test battery with 10 cognitive tests and two standard personality tests, we compared cognitive abilities of these two species. In a total of 1104 tests, we tested $n = 120$ GML and $n = 34$ MBML. As ecologically relevant abilities, we chose variation in exploration and neophilia, innovative propensities, persistence, learning abilities regarding associative and flexible learning using visual and spatial cues, and spatial memory. To also examine cognitive performances in tasks without obvious ecological relevance [4] (i.e. cognitive abilities that are not expected to covary with the degree of ecological specialization) we assessed variation in inhibitory control, means–end understanding and goal directedness (see electronic supplementary material for justification and predictions; figure 1). Finally, we examined whether cognitive performance across tests loads onto one common general intelligence ($G/g$-)factor [29,30].

## 2. Material and methods

### (a) Study site and periods
We conducted this study in Kirindy Forest, a dry deciduous lowland forest in central western Madagascar within a 12.500 ha forest concession operated by the Centre National de Formation, d'Etudes et de Recherche en Environnement et Foresterie (CNFEREF) Morondava. Mouse lemurs at Kirindy Forest have been captured on a monthly basis as part of an ongoing long-term project [22,31]. We captured GML ($n = 120$) in a population that has been regularly monitored since 1993 [32] and MBML ($n = 34$) in another population that has been monitored since 2002 [22] (see electronic supplementary material for the details of the capture procedure). Between 2017 and 2019, we conducted experiments with wild animals in temporary short-term captivity across three field seasons covering the transitions from the wet to the dry season (March–May/June) and the transitions from the dry to the wet season (August–October/November), respectively.

### (b) Study animals: housing and experimental test battery
In the following, we briefly describe the experimental procedure and the general statistical analyses. Detailed information about sample sizes, experimental set-ups, statistical analyses and repeatability analyses are provided in the electronic supplementary material.

At the field station, individually marked mouse lemurs were housed in cages of 80 cm × 80 cm × 80 cm equipped with a nestbox, several branches, an experimental platform and ad libitum access to water. We kept animals for a maximum of three ($n = 488$; in 65 cases four, in 17 cases five) nights, after which they were released at dusk at their site of capture. In total, we tested up to 150 mouse lemurs per task in a total of 1104 tests. Sample sizes differ between tasks as it was not possible to recapture all individuals until they have participated in all tasks of the test battery (electronic supplementary material, table S1).

Testing started between 18.00 and 19.00 h under red light conditions, when subjects were active and motivated, and ended when the motivation of the animals decreased. The experimental test battery comprised two personality tests, an open field test and a novel object test, and 10 cognitive tests (food extraction task, persistence test, discrimination and reversal learning paradigms with visual discrimination, visual reversal learning, spatial discrimination and spatial reversal learning, plus maze, cylinder test, two string-pulling tasks), for which we used small pieces of banana as food rewards (electronic supplementary material, table S1; figure 1).

#### (i) Personality tests
We assessed an individual's explorative tendencies in an unknown environment, using an open field test (figure 1*a*). After subjects entered the arena voluntarily, they were observed for 5 min exploring the arena. We used the duration the subjects spent locomoting as measure for *exploration* (electronic supplementary material, table S2). To assess an individual's neophilic tendencies, we introduced a novel object (figure 1*c*) directly after each open field test into the arena. We counted the number of contacts over the course of a 5 min test duration and used this contact frequency as our measure for *neophilia*.

#### (ii) Food extraction task
To assess *innovative propensities*, individuals could extract up to six food rewards from a problem-solving box with six uniform wells (figure 1*d*) within a test duration of 20 min. As initial *innovation speed*, we measured the success latency as the time span between the response (i.e. entering the experimental platform and visualizing the task) and the first success (i.e. extracting the first piece of banana; solver). In case an individual did not succeed at all (non-solver), we set its success latency to 20 min as the maximum test duration. Additionally, we counted the number of successes as measure for repeated *innovative propensity*.

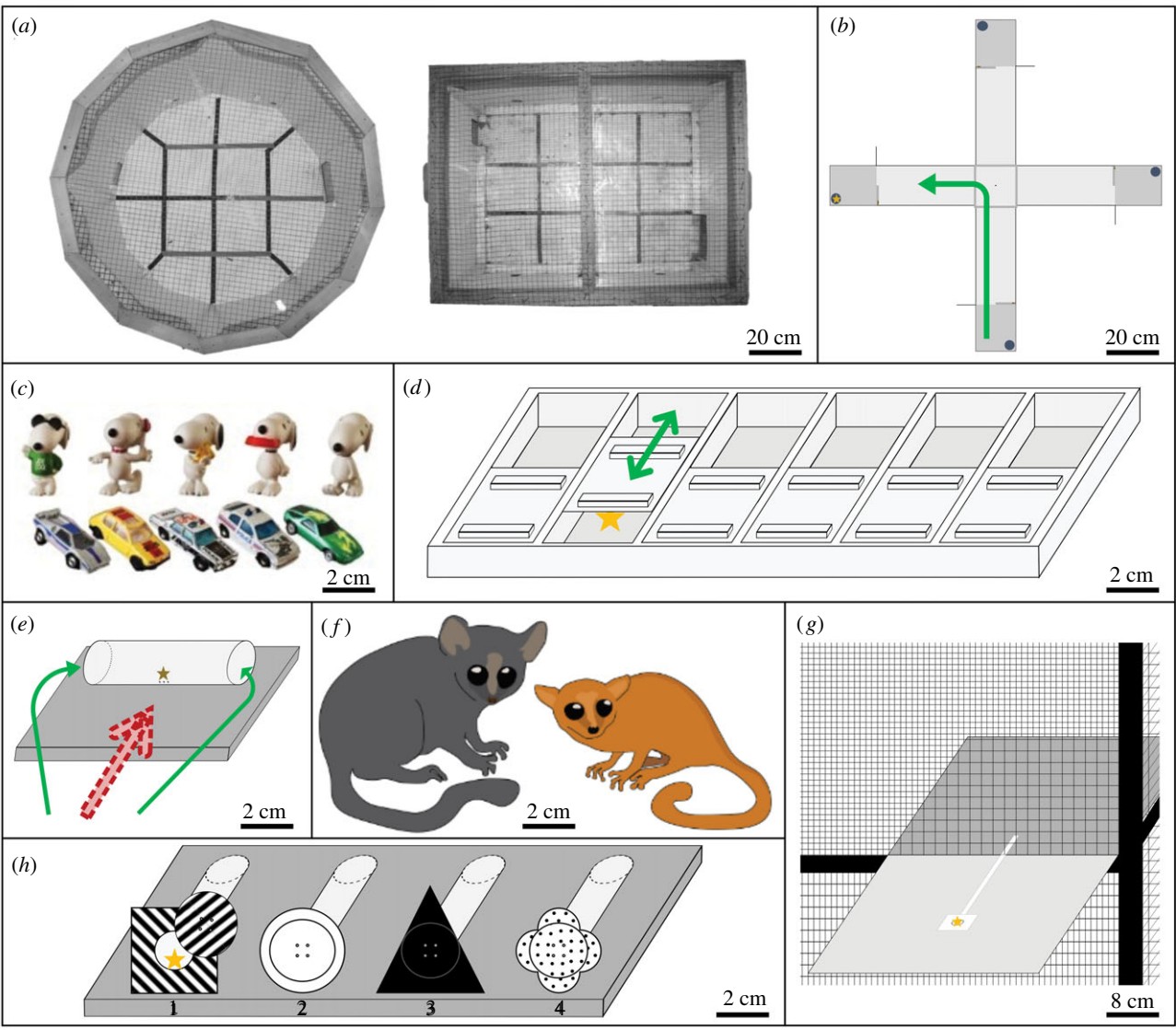

**Figure 1.** Experimental test battery and mouse lemurs. (*a*) Arenas used in the open field test. (*b*) Plus maze. (*c*) Objects used in the novel object test. (*d*) Food extraction task and persistence test. (*e*) Cylinder test. (*f*) Left: grey mouse lemur; right: Madame Berthe's mouse lemur; scaled to size differences. (*g*) String-pulling task, single-string set-up. (*h*) Apparatus used for the visual and spatial discrimination and reversal learning paradigm; numbers indicate the position of the forms. Green (filled) arrows indicate correct routes. Red (striped) arrows indicate incorrect routes. Yellow stars represent a food reward. (Online version in colour.)

### (iii) Persistence test

To assess an individual's *persistence* in manipulating an object with potential access to food, we modified the problem-solving box, in that five of the six lids were blocked and only one of the six rewards could be extracted. We calculated an individual's *persistence* rate by dividing the duration manipulating the box by the duration being in contact with the box and noted whether an individual opened the well (solver) or not (non-solver).

### (iv) Visual and spatial discrimination and repeated reversal learning paradigm

To assess an individual's *associative and flexible learning abilities*, we used a repeated discrimination and reversal learning paradigm with four separate tasks. On a plate, we positioned four tubes that only differed in shape and pattern of a form at the front part (figure 1*h*). Attached to the form was a lid that could be easily rotated to obtain access to the food reward in case of the S⁺. For the first task, the visual discrimination, and the second task, the visual reversal, the shape and pattern of the form served as cue to locate the S⁺. In the third task, the spatial discrimination and the fourth task, the spatial reversal, the shape and pattern became irrelevant, and the position of the form served as S⁺. Across sessions of 15 trials, we counted

the number of correct trials (i.e. manipulating only the S⁺ form and extracting the food reward) which we used as measure for *associative learning abilities*. After the subject had correctly chosen the S⁺-form for at least 24 out of 30 trials (80% learning criterion over two consecutive sessions), we proceeded with the next experimental task and used the total number of trials to reach this learning criterion as measure for the *overall performance* per task. As measure of *flexibility*, we calculated a transfer index (TI, equation (2.1)) for the transitions with changing reward contingencies (i.e. from the visual discrimination to the visual reversal, from the visual reversal to the spatial discrimination and from the spatial discrimination to the spatial reversal).

$$\mathrm{TI} = \frac{\text{post-reversal performance}}{\text{pre-reversal performance}}. \tag{2.1}$$

### (v) Plus maze

To assess an individual's *spatial learning abilities* and *spatial memory*, we set up a plus maze with four arms leading to four terminal boxes (figure 1*b*), of which only one was baited (goal box). We counted how often a subject entered the wrong arm and/or box per trial and summed it up to an error score per trial, which we defined as *spatial learning performance*. For the *overall spatial performance*, we used the mean sum of errors throughout one session of 15 trials.

### (vi) Cylinder test

To assess an individual's *inhibitory control*, we conducted a detour-reaching task using the cylinder test design [20] (figure 1*e*). After an initial training session with an opaque cylinder (see electronic supplementary material), we conducted the experimental session using a transparent cylinder. Throughout one session of 10 trials, we counted the number of incorrect trials (i.e. when the subject did not take the detour as a first response to get access to the food reward in the centre of the transparent cylinder).

### (vii) String-pulling task, single-string set-up

To assess an individual's *means–end understanding*, we conducted a string-pulling task in the single-string set-up (figure 1*g*). Within a 20 min test duration, the subject could pull a cable tie to access the food reward at the outer end of the cable tie. We measured the success latency as timespan between the response and reaching the reward. For subjects that did not succeed (non-solver), we set the success latency to the maximum time of the trial (20 min) plus the response latency. We used this success latency as proxy for means–end understanding.

### (viii) String-pulling task, perpendicular strings set-up

To assess an individual's *goal directedness*, we conducted a string-pulling task with a perpendicular strings set-up, as modification of the single-string set-up, where a second, non-baited cable tie was presented. In a session of 10 trials, we counted the number of incorrect trials (i.e. the subject did not succeed or it manipulated the incorrect string), which we used as proxy for *goal directedness*.

## (c) Statistical analyses
### (i) Variation in cognitive performance

We conducted all statistical analyses in R (v. 4.0.0, R Core Team, 2020), using multivariate (mixed) models to examine interspecific and intraspecific variation in performances (Gaussian linear models (LM), Gaussian linear mixed models (LMM), negative binomial models (NBM), negative binomial mixed models (NBMM), zero-inflated negative binomial models (0-infl NBM), cox-proportional hazards models (cox PHM), poisson models (PM)) and factor analytical approaches to examine general intelligence factors. Since there is no sexual dimorphism in either species, but body mass changes occur as a result of sex-specific energy strategies, as well as with hormonally induced somatic changes [22,32], we controlled for sex and body condition using the body mass index (BMI) as a proxy. In addition, we controlled for stable individual differences in behaviour (i.e. personality traits), since an individual's exploration level or neophilic tendencies can potentially influence its engagement in experimental tasks and, subsequently, its performance level. In principle, we examined interspecific variation in performances by setting species, sex and age (log-transformed) as fixed factors. To examine intraspecific variation in performance, we set sex, season, BMI (log-transformed) and personality factors as fixed factors. We tested for interactions of species and sex with other fixed factors but included the interaction only if the model significantly differed from the model without interactions. To test the significance of the predictors as a whole, we compared all full models with the respective null model comprising only the intercept and potential random factors (see electronic supplementary material) [33].

### (ii) General intelligence

Finally, we investigated general intelligence across, as well as within species. For the interspecific *G*-factor, we calculated two principal axis factor analyses, using the function 'fa' with the argument 'fm' set to 'pa' ('psych' package). The first PAF contained performance scores of individuals that completed all tests ($n = 20$ GML, $n = 9$ MBML). For the second PAF, we used performance scores of

individuals that completed all tests, except for the discrimination and reversal learning paradigm, resulting in a larger sample size ($n = 76$ GML, $n = 19$ MBML). We controlled for sphericity by applying Bartlett's test and for sampling adequacy by applying the KMO.

For the intraspecific *g*-factor, we used the same (log-transformed) performance scores as for the *G*-factor. For each species separately, we calculated two PCAs per species. The first PCA per species contained the performance scores of all tests excluding the spatial discrimination and the spatial reversal, which reduced the datasets to $n = 21$ GML and $n = 9$ MBML. For the second PCA per species, we excluded all performance scores of the repeated discrimination and reversal learning paradigm, achieving a sample size of $n = 76$ GML and $n = 19$ MBML. For each PCA, we controlled for sphericity by applying Bartlett's test and for sampling adequacy by applying the KMO.

## 3. Results

### (a) Interspecific comparisons
### (i) Personality: open field and novel object test

Since locomotion loaded most strongly on the first principal component and was most repeatable (electronic supplementary material, table S4), we retained this variable as personality trait *exploration*. Variation in *exploration* was predicted by an interaction between species and sex (LM: $p = 0.018$; electronic supplementary material, table S5, model a). Female MBML were more explorative than males as well as GML (figure 2*a*). Age did not predict variation in *exploration*.

In the novel object test, approach speed and contact frequency were poorly repeatable (approach speed: ICC = 0.158; contact frequency: ICC = 0.106). Since approach speed was skewed towards individuals that did not contact the novel object, we retained contact frequency as a measure of *neophilia*. About one-third of the individuals of both species ($n = 28$ out of 90 GML and $n = 8$ out of 24 MBML) did not contact the novel object. The full model estimating variation in *neophilia* did not significantly differ from the null model (0-infl NBM: $p = 0.073$, electronic supplementary material, table S5, model: b). Thus, variation in *neophilia* was predicted by none of species ($p = 0.489$), sex ($p = 0.791$) nor age ($p = 0.621$).

### (ii) Food extraction task: problem solving

Variation in *innovation speed* (latency to first success) differed between species (cox PHM: $p = 0.013$) and sexes ($p = 0.044$; electronic supplementary material, table S6, model a). GML were faster to extract the first food reward than MBML, and males of both species were faster than females. Age did not predict *innovation speed* ($p = 0.209$).

Variation in *innovative propensity* ($n$ opened wells) differed between species (electronic supplementary material, table S6, model b). GML opened more wells than MBML (PM: $p = 0.006$; figure 2*b*). Sex ($p = 0.670$) and age ($p = 0.874$) did not predict *innovative propensity* in either species. Both measures were repeatable (innovation speed: ICC = 0.605, innovative propensities: ICC = 0.405; $n = 21$ individuals: 15 GML and 6 MBML).

### (iii) Persistence test

Variation in *persistence* was predicted by species (LM: $p < 0.001$; electronic supplementary material, table S7, model a). GML were more persistent than MBML (figure 2*c*). Sex ($p = 0.198$) and age ($p = 0.090$) did not predict *persistence* in either species. Persistence was repeatable (ICC = 0.725,

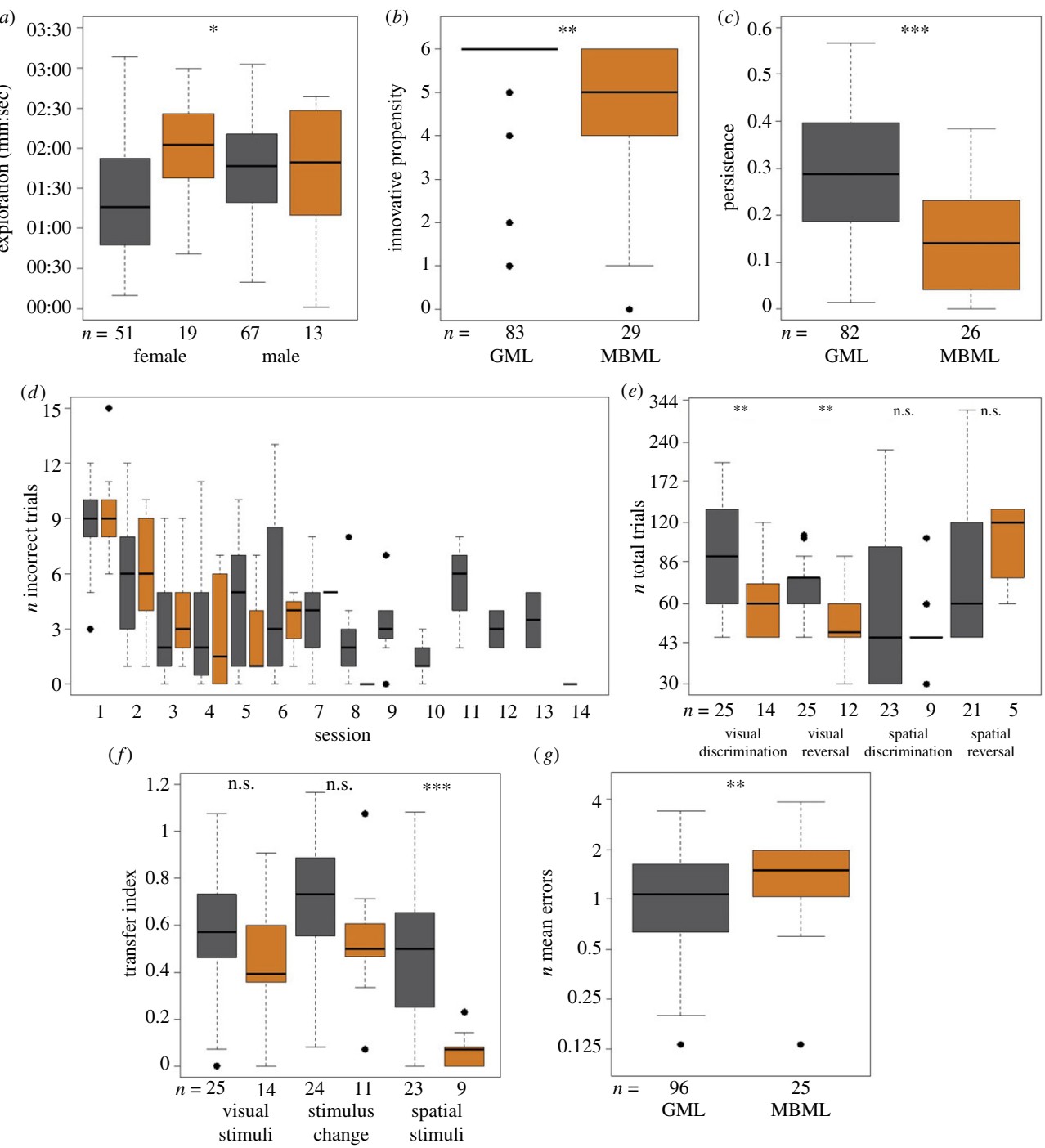

**Figure 2.** (*a*) Open field test: time spent exploring for females and males. (*b*) Food extraction task: innovative propensity (N opened wells) per species. (*c*) Persistence test: duration spent manipulating the box as a rate to the duration spent in contact with the box per species. (*d*) Visual discrimination learning across sessions per species (*p* < 0.05). (*e*) Visual discrimination and reversal learning paradigm: overall performances within tasks. (*f*) Visual discrimination and reversal learning paradigm: flexible learning between tasks. (*g*) Plus maze: spatial memory performance per species. Sample sizes (*n*) are given below each boxplot. Dark grey: GML; light grey or orange: MBML. Significance levels: \**p* < 0.05, \*\**p* < 0.01, \*\*\**p* < 0.001. (Online version in colour.)

*n* = 15 individuals, 9 GML and 6 MBML). Persistence across the two food extraction tests correlated positively (Spearman rho = 0.382, *p* < 0.001, *n* = 108) and was repeatable (ICC = 0.394).

### (iv) Visual discrimination

Variation in *visual discrimination learning* was predicted by species in an interaction with session (NBMM: *p* = 0.020) and age (*p* = 0.015; electronic supplementary material, table S8, model a). Both species decreased the number of incorrect trials across sessions, but this decrease was less pronounced in GML than in MBML (figure 2*d*). In both species, older individuals had more incorrect trials across sessions than younger individuals. Sex did not predict variation in *visual discrimination learning* across sessions (*p* = 0.102). Variation in the *overall visual discrimination performance* was predicted by species (LM: *p* = 0.009; electronic supplementary material, table S8, model b; figure 2*e*) and age (*p* = 0.008). GML and older individuals reached the learning criterion after more trials than MBML and younger individuals, respectively. Sex did not predict variation in *visual discrimination performance* (*p* = 0.458).

### (v) Visual reversal learning

Variation in *visual reversal learning* was predicted by species (NBMM: $p = 0.001$), session ($p < 0.001$) and age ($p = 0.018$; electronic supplementary material, table S9, model a). Both species decreased the number of incorrect trials across sessions. However, this decrease was less pronounced in GML than in MBML. Older individuals had fewer incorrect trials across sessions than younger individuals. Sex did not predict variation in *visual reversal learning* across sessions ($p = 0.758$). Variation in the *overall visual reversal performance* was predicted by species (LM: $p = 0.011$; electronic supplementary material, table S9, model b; figure 2e). GML reached the learning criterion after more trials than MBML ($p = 0.003$). Sex ($p = 0.614$) and age ($p = 0.186$) did not predict variation in the *visual reversal performance*.

Neither species (LM: $p = 0.310$; figure 2f), sex ($p = 0.192$) nor age ($p = 0.054$; electronic supplementary material, table S9, model c) had a significant effect on the transfer index (TI) (i.e. *flexible learning* from visual discrimination to visual reversal learning). The TI did not correlate with the number of trials to reach the learning criterion in the visual reversal learning (Spearman rho = 0.006, $p = 0.974$, $n = 36$) or with the *innovative propensity* (Spearman rho = 0.144, $p = 0.401$, $n = 36$).

### (vi) Spatial discrimination

Variation in *spatial discrimination learning* was predicted by species in an interaction with session (NBMM: $p = 0.002$), sex in an interaction with session ($p = 0.001$), age in an interaction with species ($p = 0.005$), and by the position of the rewarded tube $S^+$ ($p < 0.001$; electronic supplementary material, table S10, model a). Both species and sexes decreased the number of incorrect trials across sessions. However, in GML and females this decrease was less pronounced than in MBML and males, respectively. While older GML had more incorrect trials across sessions than younger individuals, older MBML had fewer incorrect trials than younger ones. Mouse lemurs that learned to associate position 1 ($S^+$) with the food reward had fewer incorrect trials than individuals learning the positions 2, 3 or 4. Variation in *spatial discrimination performance* was predicted by species (figure 2e) in an interaction with age (LM: $p = 0.043$), and by the $S^+$ ($p = 0.002$; electronic supplementary material, table S10, model b). While older GML reached the learning criterion after more trials than younger individuals, the effect was reversed for MBML. Position 1 was learned after fewer trials than the other three positions. Sex ($p = 0.931$) did not predict variation in *spatial discrimination performance*.

The TI from visual reversal learning to spatial discrimination was predicted by sex (LM: $p = 0.014$), with males achieving a higher TI than females, but not by species ($p = 0.066$; figure 2f) and age ($p = 0.680$; electronic supplementary material, table S10, model c) The TI correlated negatively with the number of trials to reach learning criterion in the spatial discrimination task (Spearman rho = −0.463, $p = 0.008$, $n = 32$). Hence, initially more flexible individuals (higher TI) were faster learners (fewer trials to reach the learning criterion). However, TI did not correlate with *innovative propensity* (Spearman rho = 0.039, $p = 0.831$, $n = 33$).

### (vii) Spatial reversal learning

Variation in spatial reversal learning was predicted by species in interaction with session (NBMM: $p = 0.004$), sex ($p = 0.034$) and by the $S^+$ ($p = 0.041$; electronic supplementary material, table S11, model a). In both species, the number of incorrect trials decreased across sessions. However, in GLM this decrease was less pronounced than in MBML. Females had fewer incorrect trials across sessions than males. Mouse lemurs that learned to associate position 1 with the food reward ($n = 7$), had fewer incorrect trials than others (position 2: $n = 5$, position 3: $n = 6$, position 4: $n = 8$). Age did not predict variation in *spatial reversal learning* ($p = 0.931$). Variation in the *overall spatial reversal performance* was predicted by sex in interaction with the $S^+$ (LM: $p = 0.012$, electronic supplementary material, table S11, model b). Male mouse lemurs that learned to associate position 3 with the food reward ($n = 3$), reached the learning criterion after more trials than the others. Species ($p = 0.267$; figure 2e) and age ($p = 0.122$) did not predict variation in the *spatial reversal performance*.

The TI from spatial discrimination to spatial reversal learning was predicted by species (LM: $p < 0.001$), with GML achieving a higher TI than MBML (electronic supplementary material, table S11, model c; figure 2f). Sex ($p = 0.168$) and age ($p = 0.492$) did not predict variation in *flexible learning*. TI correlated negatively with the number of trials to reach the learning criterion in the spatial reversal learning task (Spearman rho = −0.522, $p = 0.006$, $n = 26$). Hence, initially more flexible individuals (higher TI) were faster learners (fewer trials to reach the learning criterion). However, TI did not correlate with *innovative propensity* (Spearman rho = 0.112, $p = 0.549$, $n = 31$). Since we only tested one to two GML repeatedly in 2–4 of the tasks, we could not estimate repeatability of the performance scores in this experiment.

### (viii) Plus maze: spatial memory

Variation in *spatial learning* was predicted by species in an interaction with trial (NBMM: $p = 0.005$) and by the goal box ($p < 0.001$; electronic supplementary material, table S12, model a). Both species made fewer errors across trials, and mouse lemurs assigned to the straight goal box ($n = 22$) made fewer errors across trials than those assigned to the left ($n = 52$) or right goal box ($n = 47$). Sex ($p = 0.254$) and age ($p = 0.537$) did not predict variation in *spatial learning* across trials.

Variation in *spatial memory* was predicted by species (LM: $p = 0.003$) and by the goal box ($p < 0.001$, electronic supplementary material, table S12, model b). GML made fewer errors than MBML (figure 2g). Mouse lemurs assigned to the left or right goal box made more errors than those assigned to the straight goal box. Sex ($p = 0.361$) and age ($p = 0.429$) did not predict variation in *spatial memory*. *Spatial memory* was repeatable ICC = 0.412 ($n = 21$ individuals, 15 GML and six MBML).

### (ix) Cylinder test: inhibitory control

In the inhibitory control task, individuals that needed more trials to reach the learning criterion prior to testing made more errors during the testing session (NBM: $p = 0.005$). Variation in *inhibitory control* was not predicted by either species ($p = 0.126$), sex ($p = 0.783$) or age ($p = 0.319$; electronic supplementary material: electronic supplementary material table S13, model b).

*Inhibitory control* did not correlate with *innovative propensity* (Spearman rho = −0.152, $p = 0.121$, $n = 105$), *flexible learning*

(TI from visual reversal learning: Spearman rho = −0.096, $p = 0.576$, $n = 36$; TI visual reversal to spatial discrimination learning: Spearman rho = −0.030, $p = 0.869$, $n = 36$; TI from spatial reversal learning: Spearman rho = −0.026, $p = 0.891$, $n = 33$) or *overall learning performances* (visual discrimination: Spearman rho = −0.133, $p = 0.439$, $n = 36$; visual reversal: Spearman rho = −0.007, $p = 0.970$, $n = 34$; spatial discrimination: Spearman rho = 0.008, $p = 0.966$, $n = 31$; spatial reversal: Spearman rho = 0.078, $p = 0.713$, $n = 25$). The number of training trials (ICC = 0.556) were repeatable, but the number incorrect test trials were only poorly repeatable (ICC = 0.141).

### (x) String-pulling task: means–end understanding and goal directedness

Performance in the *means–end understanding* (cox PHM: $p = 0.600$) and *goal directedness* (PM: $p = 0.874$) was not predicted by any of the investigated factors (electronic supplementary material, table S14, models a, b). We estimated the repeatability for $n = 11$ individuals (7 GML and 4 MBML) that repeated this test (response latency: ICC = −0.137, success latency: ICC = −0.192).

### (b) General intelligence

We did not find evidence for an interspecific *G*-factor in either mouse lemur species. The performance scores did not load similarly onto the first component of the PAF, including data from the larger sample size (including tests on problem solving, spatial memory, means–end understanding, goal directedness, inhibitory control), and Bartlett's tests of sphericity was non-significant, indicating a generally low correlation across performance scores (electronic supplementary material, table S15). The results were similar for the reduced PAF (including tests on visual and spatial discrimination as well as reversal learning, electronic supplementary material, table S16).

### (c) Intraspecific variation In performance in personality and cognitive tests

We also investigated whether intraspecific performance in these tests was influenced by sex, age, BMI or in case of cognitive tests, also by variation in personality traits. In both species, variation in performance scores was only occasionally explained by these individual characteristics, with no systematic variation within species. We also did not find evidence for an intraspecific *g*-factor. The complete results and the discussion thereof can be found in the electronic supplementary material (GML, table S17; MBML, table S18).

## 4. Discussion

Grey and Madame Berthe's mouse lemurs differed in 13 of 21 performance measures (table 1), which were moderately repeatable in the majority of tests and hence consistent within individuals (electronic supplementary material, table S2). Overall, the ecological generalist GML were more innovative, more persistent, showed better spatial learning and memory, and greater flexibility when spatial stimulus–reward contingencies were changed. The ecologically more specialized MBML were more explorative and learned visual and spatial reward contingencies faster, achieving better total performances in visual association learning. However, the two species did not differ in neophilia, inhibitory control, means–end understanding or goal directedness. Moreover, we did not find evidence for an interspecific *G*- or intraspecific *g*-factor. In summary, our study provides support for domain-specific cognitive coevolution with ecological factors, and a particular advantage of generalists in confronting novel challenges with a greater innovative potential and greater flexibility when confronted with changing spatial stimuli.

In both species, performance in personality and cognitive tests was weakly and inconsistently influenced by individual characteristics and moderately repeatable, reflecting general patterns on intra-individual consistency in cognitive performances in animals [34]. Regarding personality traits, both species did not differ in exploration. However, female MBML were more explorative than males in comparison to GML. In addition, the two species did not differ in their neophilic response, supporting neither the dangerous-niche hypothesis (i.e. generalists should be more neophobic as they may encounter more dangerous situations) nor the neophobia threshold hypothesis (i.e. generalists should be less neophobic as they may have more diverse prior experiences; [34,35]).

Concerning cognitive performance, GML were indeed more innovative and achieved higher flexibility scores, at least under changing spatial stimuli. These abilities may allow them to respond more adaptively to dynamically changing habitats and anthropogenic influences [26,37]. In our study region, the abundance of GML is generally higher than that of MBML, especially in habitats with anthropogenic influence, such as edge habitats [26,37]. Similarly, in northern Madagascar, GML were largely unaffected by habitat fragmentation, while the abundance of the sympatric but ecologically more specialized golden-brown mouse lemurs (*M. ravelobensis*) decreased with increasing habitat fragmentation [25]. Increasing anthropogenic activities, such as deforestation or habitat fragmentation, contribute to an alarming species loss in Madagascar [38]. Such environmental changes may eventually promote a species turnover towards ecological generalists, whereas more specialized species may suffer from decreased population size [28,39]. Our study therefore indicates potential cognitive constraints of ecological specialization that may help to explain why some species experience a higher extinction risk in the face of ongoing environmental changes.

MBML learned the associative reward contingencies faster; a characteristic that allows specialists to experience advantages over generalists in stable environmental conditions [40]. Specifically, MBML's better motor control to abandon previously successful behaviors [41] and their lower persistence to produce a behaviour that does not lead to success (this study), may have contributed to their superior performance, at least in visual associative learning experiments. GML feed relatively more often on tree gum, a foraging strategy associated with enhanced inhibitory control [42]. However, the two species did not differ in inhibitory control, suggesting that the fact that they feed on gum rather than the relative frequency of this foraging behaviour is associated with superior inhibitory control. Both inhibitory control and reversal learning abilities reflect behavioural flexibility [20,43]. However, these measures did not correlate with each other, suggesting that they reflect different aspects of cognitive flexibility in mouse lemurs. The TI is a standard proxy for flexible learning abilities, reflecting the potential to switch between tactics when conditions change (i.e. in the context of a first response to a modified reward contingency)

**Table 1.** Overview of species differences in performance across tasks. (↑) indicates better and (↓) indicates worse performance, whereas (—) indicates no difference in performance.

| task | GML | MBML | age | sex |
|---|---|---|---|---|
| activity | — | — | — | ↓ GML: females |
| neophilia | — | — | — | ↓ males |
| innovation: | | | | |
| speed | ↑ | ↓ | — | ↑ males |
| propensity | ↑ | ↓ | — | — |
| persistence | ↑ | ↓ | — | — |
| visual discrimination: | | | | |
| learning | ↓ | ↑ | ↓ old | — |
| performance | ↓ | ↑ | ↓ old | — |
| visual reversal: | | | | |
| learning | ↓ | ↑ | ↑ old | — |
| performance | ↓ | ↑ | — | — |
| spatial discrimination: | | | | |
| learning | ↓ | ↑ | ↓ old GML ↑ old MBML | ↑ males |
| performance | — | — | ↓ old GML, ↑ old MBML | — |
| spatial reversal: | | | | |
| learning | ↓ | ↑ | — | ↓ males |
| performance | — | — | — | ↓ males: position 3 |
| flexibility (TI) | | | | |
| visual | — | — | — | — |
| visual-spatial | — | — | — | ↑ males |
| spatial | ↑ | ↓ | — | — |
| spatial memory: | | | | |
| learning | ↑ | ↓ | — | — |
| performance | ↑ | ↓ | — | — |
| inhibitory control | — | — | — | — |
| means–end understanding | — | — | — | — |
| goal directedness | — | — | — | — |

[44]. However, it does not indicate how well the old tactic will be abandoned in favour of the new tactic. This might be better reflected by the number of trials until criterion after reversal or the number of perseverative errors across sessions, because these measures better reflect how quickly individuals may overcome the previously learned reward contingency and, therefore, how flexible an animal switches between strategies [45]. The TI and the number of trials to reach the criterion after reversal correlated negatively with each other in the transfer from the visual to spatial stimuli and in the spatial reversal learning, indicating that individuals that responded more flexibly to the reversed reward contingency learned this reward contingency faster in spatial learning. Hence, with regard to spatial stimuli, which might be ecologically more relevant than abstract forms as in the visual reversal paradigm, mouse lemurs were able to switch flexibly between strategies, and GML were more flexible when confronted with changing spatial stimuli than MBML.

The plus maze assays spatial learning and memory that is essential for effective spatial navigation. GML learned this task faster than MBML, supporting earlier results on their spatial learning and high travel efficiency [46]. Although MBML have larger home ranges and should therefore have better spatial abilities, the ability to adapt to different habitat types may require more flexible spatial learning abilities and may therefore better explain from an evolutionary perspective why GML performed better in this task and also why they responded more flexibly when spatial stimulus–reward contingencies changed.

In tasks with little ecological relevance, where species differences in cognitive performance were not predicted regarding the degree of ecological specialization, such as inhibitory control, means–end understanding or goal directedness, both species performed on par, suggesting that they did not differ *per se* in cognitive performance. These results also support our interpretation of the adaptive nature of the observed differences as being driven by different ecological factors.

In humans, cognition has evolved towards positively correlating generally high-level cognitive abilities [47,48]. However, a general correlation between cognitive abilities and brain size has received mixed support in comparative studies of animals [47]. In mouse lemurs, we did not find

evidence for general intelligence, neither on the interspecific, nor on the intraspecific level. Our results rather support the domain-specific hypothesis [49], which postulates enhanced abilities in only some cognitive domains, whereas abilities in others remain on more basic performance levels [50].

Finally, this study raises new questions about the evolutionary mechanisms driving cognitive adaptations to environmental features. From a phylogenetic perspective, the split between the basic lineages to which these two species belong occurred about 8–10 Ma. Phylogenetic reconstructions of the speciation patterns in mouse lemurs suggests that the longitudinal dispersal along the west coast of Madagascar by GML was achieved with relative ease throughout the Pleistocene [21]. However, habitat fragmentation via Holocene droughts may have erected natural barriers such as rivers, creating several centres of endemism [51], isolating some species, such as MBML, in small ranges. Thus, GML actually had more time available to evolve cognitive and ecological adaptations to the habitat in which they now co-occur, but they appear to have retained cognitive abilities that may provide ecological advantages across their entire range. Genetic studies investigating patterns of gene flow and heritability in different cognitive abilities are now indicated to begin exploring the evolutionary mechanisms shaping the links between ecological adaptations and cognitive constraints.

## 5. Conclusion

We show that direct comparisons of cognitive performances between sympatric sister species with a similar social system can help to unfold the role of ecological factors in the evolution of cognition. Species-specific ecological adaptations covary with cognitive abilities. The ecologically more generalist species was particularly more innovative, persistent, exhibited better spatial learning abilities, spatial memory, as well as spatial flexibility than the specialist, affording them with the behavioural flexibility to respond adaptively to rapidly changing habitats and anthropogenic disturbances. How these differences in cognitive abilities have been maintained over millions of generations in local sympatry requires further study.

Data accessibility. The data are provided in electronic supplementary material [52].

Authors' contributions. J.H.M.: conceptualization, data curation, formal analysis, investigation, methodology, project administration, visualization, writing-original draft; P.M.K.: conceptualization, funding acquisition, methodology, project administration, resources, supervision, writing-review and editing; C.F.: conceptualization, funding acquisition, methodology, project administration, supervision, writing-original draft, writing-review and editing.

All authors gave final approval for publication and agreed to be held accountable for the work performed therein.

Competing interests. The authors declare no competing interests.

Funding. This research was funded by the Deutsche Forschungsgemeinschaft (DFG), awarded to P.M.K. (grant no. KA 1082/29-565 2) and C.F. (grant no. FI 929/10-1).

Acknowledgements. We are grateful to Bruno Tsiverymana and the other members of the Equipe Kirindy for their help with capturing and housing mouse lemurs, and to Jan Förster, Joe Buchner, Luca Hahn, Svenja Blödorn, Joana Niedner, Colleen Illing, Annkatrin Pahl, Frederike Hoppmann and Tímea Kovács for their help with collecting data and analysing videos. We thank the Ministry of the Environment, the Mention Zoologie et Biodiversité Animale Université d'Antananarivo and the CNFEREF Morondava for their authorization and support of this study, Katja Rudolph for providing mouse lemur icons, and two referees for very constructive comments on an earlier version of this manuscript.

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
