## [Peer Review File · Proceedings of the Royal Society B: Biological Sciences]

Review History

RSPB-2021-1728.R0 (Original submission)

Review form: Reviewer 1 (Albert Phillimore)

Recommendation

Accept with minor revision (please list in comments)

Scientific importance: Is the manuscript an original and important contribution to its field?

Good

General interest: Is the paper of sufficient general interest?

Good

Quality of the paper: Is the overall quality of the paper suitable?

Good

Is the length of the paper justified?

Yes

Should the paper be seen by a specialist statistical reviewer?

No

Do you have any concerns about statistical analyses in this paper? If so, please specify them explicitly in your report.

No

It is a condition of publication that authors make their supporting data, code and materials available - either as supplementary material or hosted in an external repository. Please rate, if applicable, the supporting data on the following criteria.

Is it accessible?

Yes

Is it clear?

Yes

Is it adequate?

Yes

Do you have any ethical concerns with this paper?

No

Comments to the Author

This manuscript evaluates species differences on a battery of cognitive tasks implemented with two sympatric mouse lemur species. The main question is whether variation in ecological features (generalist versus specialist) explains differences in cognitive abilities between these species. Results showed that the more generalist species exhibited better spatial learning and innovated more than the more specialist species. In general, I found this study to be well-designed, and analyzed appropriately, with results that can contribute to our understanding of cognitive evolution. However, I have few comments for the authors' consideration.

The introduction on Line 83-86 reports the logic behind the choice of the cognitive tasks administered. It is not immediately clear what the authors mean when they say that inhibitory control, means-end understanding, and goal-directedness are not obvious ecological relevant. Did the authors mean that these abilities can also be used in contexts involving social rather than feeding decisions? A similar argument could be made for the other kind of tasks since associative and flexible learning or spatial memory could also be used in social situations. Perhaps this could be clarified.

From the method and the Supplementary materials is not clear whether each subject received all the tasks of the battery or different subjects were tested in different tasks (between-subjects procedure). Please clarify.

To calculate the interspecific G-factor, the authors implemented principal axis factor analyses, while to calculate the intra-specific g-factor, they implemented PCAs. What is the logic behind this choice?

Lastly, I found some results and, consequently, some parts of the discussion unclear. In particular, from the results section and Table S20 seems that the grey mouse lemurs perform better than the Mme. Berthe's mouse lemurs only in the tasks measuring innovation, persistence, and spatial memory, but not in those measuring visual and spatial discrimination and reversal. Thus, it appears that, although less innovative, MBML are quite flexible. Lines 393-397 report that MBML's relatively low persistence contributes to their higher associative learning abilities. Yet, they performed better than the other species also when the contingencies were switched. Thus, this result does not fully support the hypothesis that generalists species might have greater flexible learning abilities than more specialist species. Some discussion of this pattern of results will help clarify the paper's main conclusion.

Review form: Reviewer 2

Recommendation

Accept with minor revision (please list in comments)

Scientific importance: Is the manuscript an original and important contribution to its field?

Excellent

General interest: Is the paper of sufficient general interest?

Excellent

Quality of the paper: Is the overall quality of the paper suitable?

Excellent

Is the length of the paper justified?

No

Should the paper be seen by a specialist statistical reviewer?

Yes

Do you have any concerns about statistical analyses in this paper? If so, please specify them explicitly in your report.

Yes

It is a condition of publication that authors make their supporting data, code and materials available - either as supplementary material or hosted in an external repository. Please rate, if applicable, the supporting data on the following criteria.

Is it accessible?

Yes

Is it clear?

Yes

Is it adequate?

No

Do you have any ethical concerns with this paper?

No

Comments to the Author

Comments to authors

Summary

This manuscript describes a study looking at cognition and personality across two mouse lemur species. The study spans multiple field seasons, high samples sizes (at least in one species) and uses a large number of cognitive tests supplemented by a few personality tests. I want to commend the authors for this great study. It is quite extensive and will be a great addition to the literature. I found the manuscript well written and therefore easy to follow. I have a few comments and questions regarding the main manuscript and the supplementary which details most of the testing procedures and statistical analyses.

Title

I think the title is catchy but also an overstatement. "Evolution of primate cognition" would fit better as the title for a big comparative study looking at many primate species. In your study you

only looked at 2. This means your sample size in regards to your statement on the evolution of primate cognition is 2 and insufficient. I suggest rephrasing. The rest of the title is great!

Main Manuscript

L 35-42: This is a nice introductory paragraph. I am just a little worried about the generality of the statements while mostly primate literature is cited. I suggest either making the statements more primate centered or adding literature from other taxonomic groups.

L 81: You talk about 1,104 experiments here but use tests instead of experiments across the rest of the manuscript (and the supplementary material). I suggest using tests.

L 82: What you measured with your open field test is actually exploration because activity can only be measured in a familiar environment while exploration occurs in unfamiliar space. I suggest you change this in the main MS and the supplementary material.

L 112: Do you have any evidence that your two lemur species cannot perceive red light in the wavelength your light is emitting?

L 131: what was "the response" exactly?

L 139: what does "in relation" mean here exactly? Could you make this description more clear?

L 142-157: what was the inter-trial interval, the inter-session interval and how did you calculate the transfer index.

L 166: I am not sure you can call the cylinder task "classical". Just state that you followed MacLean et al.

L 198: With null model you mean an intercept only model? I think you could give this information here. I would also really appreciate if you could add a justification why you did compare full and null models and when you report results from the full models.

Results generally: I think it would be a good idea to add the type of test used to generate your results in the main manuscript. After reading the supplementary, I know you used a number of different approaches like linear mixed models or cox-hazard models depending on the data. I understand that you are limited in space and cannot present many details about all the analyses in the main MS. However, just adding the name of the test in front of your statistical estimates presented in the results text would give readers some idea about analysis. They can then consult the supplementary for more details if needed.

L 258 and following: please be aware that you used Gaussian instead of a Poisson distribution to analyse number of incorrect choices across sessions. This should be corrected.

L 325-327: I wonder if you need to add a description of the training to the methods description of the cylinder task. I am familiar with the task and know how it works but other readers might not know that there is a training phase with an opaque cylinder. This is also an important part of the test because this training ensures that what is measured in the transparent phase is actually not confounded with learning the motor response of detouring. If you have space then I would shortly describe the training as well.

L 329-336: I find this very interesting because, as far as I am aware, nobody really finds a correlation of inhibitory control across tasks. Also, I find it interesting that you have the lowest inter-observer reliability in the cylinder task (as per table S2). I have found similar low inter-observer reliability in this task because it is quite hard to score.

L 363-365 & 369: First, you describe your results in regards to which species performed better in the tasks and then you give a summary about the co-evolution of the cognitive abilities and ecological factors focusing on the hypothesis that generalists having better innovation potential and spatial flexibility. This is a little confusing because above you state that GML, the generalist of the two species, has better spatial learning abilities but do not mention flexibility. Also, stating that MBML learn associative reward contingencies faster misses the fact that this result comes from the visual task. I think this first paragraph could be revised to better describe the results. You have a lot of them and this makes it hard to write a concise but detailed summary. I have found that drawing a mental map can help with this issue.

L 372-376: Maybe exploration of novel space, which is what you measured, does not correlate with home range size. If activity is measured properly, in familiar space, the picture might look quite different.

L277-378: You do not introduce these hypothesis in the main MS but have them quite detailed in the supplementary (predictions). I suggest adding a few sentences explaining them and how your data do not support them.

L 379-392: This is quite the nice paragraph. Very nicely written.

L 393: Again, are you taking about the visual tasks specifically? Because the results from the spatial tasks don't really show this.

L 395-397: Are you referring to the visual reversal learning now? Also, how did you tests that motor control and lower persistence actually contributed to their good performance in the visual learning tasks?

L 450: I would rephrase. "elucidate" is a quite strong word for a study with a sample size of 2.

L 586: Again, what does "in relation" mean?

Figures in general: How did you decide which data to present visually and which not? I would think that this is related to the significance of the result. Why is, for example, the spatial learning not shown in which the GML was faster. Also, you could plot the reversal performance after the discrimination performance in both visual and spatial learning tasks.

Figure 1: I really like it. Very clear!

Supplementary material

The supplementary material includes many details about the testing procedures and analysis. I have therefore some questions and comments I list below:

L58-59: Could you give the age range for each species separately?

L 92: Why did you choose to only double code 10% of videos when the field standard is 20%?

L 133-146: Open field test. You describe the repeated testing of certain individuals to calculate repeatability. I miss the information on how much time (on average) passed between these repeated measures. You do state further above that lemurs were recaptured within at least 3 nights after their last release. I am just wondering because the time between measurements can greatly affect repeatability measures with short intervals often giving higher repeatability. Did you also retest individuals across years?

L 159-160: Please explain what rears and dips are.

L 161-164: What was used as the response variable "activity" in these models?

L 178-183: Again you do not mention the response variables here. I assume you ran 3 models one for each measure you describe in the paragraph above? Or did you run 4 models also looking at the approach speed? If you ran multiple test to answer the same question about neophilia, did you consider to adjust your alpha levels? I see you do that multiple times but you mostly report the results from the intercept model? Do I interpret your results tables correctly?

L 204-206: I understand that your low samples size in MBML prevented you from analysing your data using complex models. This also means, however, that you again used multiple test to basically answer the same questions. Did you consider adjusting your alpha level?

L 220-223: Are you not worried that giving 17 individuals extra training might have influenced their performance in the following persistence test?

L 233: Could you please repeat how you calculated persistence here? It is not clear from the main manuscript.

L 292: There is an error with the reference

L 298. I think this calculation should at least be mentioned in the main MS

L 306-307: Number of incorrect trials usually follows a Poisson distribution not a Gaussian distribution because it is count data. Also, associative learning ability is not mentioned in the main MS but instead you describe visual discrimination, reversal, spatial discrimination and spatial reversal learning ability. Is this equivalent?

L 356: Do I understand correctly, that you ran 15 trials per night (also in the visual and spatial discrimination and reversal learning task)? Meaning within 3 days you ran 30 trials? This is, unfortunately, not described and missing from the methods description.

L 413-414: Were these holes also present in the opaque cylinder? Generally, what experience do your animals have with transparency, as this can greatly affect performance in this task (amongst a number of other non-cognitive factors which lead to a lot of criticism of the cylinder task).

L 502: Are you sure you mean exclude here? I think you mean include, because by excluding it your sample size increases which you state in the next sentence.

L 579-581: Visual and spatial discrimination and reversal learning. I see that your sample size decreases. I am not sure if you stated this properly above but does that mean that some individuals did not reach the learning criterion or that they could not be recaptured. You might want to give some additional explanation in the table caption about this fact.

L 587-588: I think there is a repeat in the caption: between the first, second, and third test, and between the first, second, and third test.

Results Tables: I am wondering, based on how you describe it I assumed that you only presented the results for fixed effects if the comparison with the null model was significant, so they were different. But in table S5, neophilia, it is above 0.05 (same for table S13, S15). In table S18 and S19 results of fixed effects are only given if the comparison was significant. Why did you choose to not do this in the inter-species models? I might have not fully understood your selection process about what to report. I think this could be improved.

L 675: I think your arrow symbols got lost in the table caption. I see an \bar{n} and a \bar{o} in the caption but arrows in the table.

Data file

The datafile is a summary table not a raw datafile. If I understood your analyses correctly, for some models you used trial by trial data or at least session by session data which is not present in the provided datafile, or at least I think so. A readme file which explains your columns would also be great as the column names are not all self-explanatory. Finally, you missed the opportunity to provide r code that can be used to reproduce your results. Therefore, unfortunately, the reproducibility of your study is diminished.

Overall, I think this is a great study and well written manuscript. This will be of great interest to a general audience because it reveals quite interesting results in regards to the ecology of the species. I applaud the dedication of the researcher to conduct such a long and complex study with wild caught animals!

Birgit Szabo

Decision letter (RSPB-2021-1728.R0)

20-Sep-2021

Dear Dr Henke-von der Malsburg:

Your manuscript has now been peer reviewed and the reviews have been assessed by an Associate Editor. The reviewers' comments (not including confidential comments to the Editor) and the comments from the Associate Editor are included at the end of this email for your reference. As you will see, the reviewers and the Editors have raised some concerns with your manuscript and we would like to invite you to revise your manuscript to address them.

Research ethics:

Use of animals and field studies:

It is a condition of publication that you make available the data and research materials supporting the results in the article. Please see our Data Sharing Policies (<https://royalsociety.org/journals/authors/author-guidelines/#data>). Datasets should be deposited in an appropriate publicly available repository and details of the associated accession number, link or DOI to the datasets must be included in the Data Accessibility section of the article (<https://royalsociety.org/journals/ethics-policies/data-sharing-mining/>). Reference(s) to datasets should also be included in the reference list of the article with DOIs (where available).

Please submit a copy of your revised paper within three weeks. If we do not hear from you within this time your manuscript will be rejected. If you are unable to meet this deadline please let us know as soon as possible, as we may be able to grant a short extension.

Best wishes,
Dr Robert Barton
mailto:proceedingsb@royalsociety.org

Associate Editor
Board Member: 1
Comments to Author:

We have obtained two reviews of your manuscript, and I am happy to tell you that both the reviewers liked your paper. My own reading of your manuscript supports the reviews: this is an interesting piece of work that could become publishable in Proceedings B. However, although the reviews were generally very positive, one of the reviewers came up with a long list of useful suggestions and clarifying questions, which should lead to an improved manuscript. The comments by the other reviewer are fewer and perhaps easier to deal with, but they too would likely improve your paper. I am of the same impression as the reviewers that you could improve the manuscript with several changes including providing more methodological detail, justifying and explaining your statistical choices more clearly, and interpreting the somewhat conflicting results more carefully. Review 2 is correct to note that it can be difficult to summarize so many experiments both in detail and concisely but the reviewer makes some suggestions on what needs clarifying. I look forward to seeing your revised manuscript at which time I will decide if it needs to be seen by the reviewers again.

Reviewer(s)' Comments to Author:
Referee: 1

Comments to the Author(s)

This manuscript evaluates species differences on a battery of cognitive tasks implemented with two sympatric mouse lemur species. The main question is whether variation in ecological features (generalist versus specialist) explains differences in cognitive abilities between these species. Results showed that the more generalist species exhibited better spatial learning and innovated more than the more specialist species. In general, I found this study to be well-designed, and analyzed appropriately, with results that can contribute to our understanding of cognitive evolution. However, I have few comments for the authors' consideration.

The introduction on Line 83-86 reports the logic behind the choice of the cognitive tasks administered. It is not immediately clear what the authors mean when they say that inhibitory control, means-end understanding, and goal-directedness are not obvious ecological relevant. Did the authors mean that these abilities can also be used in contexts involving social rather than feeding decisions? A similar argument could be made for the other kind of tasks since associative and flexible learning or spatial memory could also be used in social situations. Perhaps this could be clarified.

From the method and the Supplementary materials is not clear whether each subject received all the tasks of the battery or different subjects were tested in different tasks (between-subjects procedure). Please clarify.

To calculate the interspecific G-factor, the authors implemented principal axis factor analyses, while to calculate the intra-specific g-factor, they implemented PCAs. What is the logic behind this choice?

Lastly, I found some results and, consequently, some parts of the discussion unclear. In particular, from the results section and Table S20 seems that the grey mouse lemurs perform better than the Mme. Berthe's mouse lemurs only in the tasks measuring innovation, persistence, and spatial memory, but not in those measuring visual and spatial discrimination and reversal. Thus, it appears that, although less innovative, MBML are quite flexible. Lines 393-397 report that MBML's relatively low persistence contributes to their higher associative learning abilities. Yet, they performed better than the other species also when the contingencies were switched. Thus, this result does not fully support the hypothesis that generalists species might have greater flexible learning abilities than more specialist species. Some discussion of this pattern of results will help clarify the paper's main conclusion.

Referee: 2

Comments to the Author(s)

Comments to authors

Summary

This manuscript describes a study looking at cognition and personality across two mouse lemur species. The study spans multiple field seasons, high samples sizes (at least in one species) and uses a large number of cognitive tests supplemented by a few personality tests. I want to commend the authors for this great study. It is quite extensive and will be a great addition to the literature. I found the manuscript well written and therefore easy to follow. I have a few comments and questions regarding the main manuscript and the supplementary which details most of the testing procedures and statistical analyses.

Title

I think the title is catchy but also an overstatement. "Evolution of primate cognition" would fit better as the title for a big comparative study looking at many primate species. In your study you only looked at 2. This means your sample size in regards to your statement on the evolution of primate cognition is 2 and insufficient. I suggest rephrasing. The rest of the title is great!

Main Manuscript

L 35-42: This is a nice introductory paragraph. I am just a little worried about the generality of the statements while mostly primate literature is cited. I suggest either making the statements more primate centered or adding literature from other taxonomic groups.

L 81: You talk about 1,104 experiments here but use tests instead of experiments across the rest of the manuscript (and the supplementary material). I suggest using tests.

L 82: What you measured with your open field test is actually exploration because activity can only be measured in a familiar environment while exploration occurs in unfamiliar space. I suggest you change this in the main MS and the supplementary material.

L 112: Do you have any evidence that your two lemur species cannot perceive red light in the wavelength your light is emitting?

L 131: what was "the response" exactly?

L 139: what does "in relation" mean here exactly? Could you make this description more clear?

L 142-157: what was the inter-trial interval, the inter-session interval and how did you calculate the transfer index.

L 166: I am not sure you can call the cylinder task "classical". Just state that you followed MacLean et al.

L 198: With null model you mean an intercept only model? I think you could give this information here. I would also really appreciate if you could add a justification why you did compare full and null models and when you report results from the full models.

Results generally: I think it would be a good idea to add the type of test used to generate your results in the main manuscript. After reading the supplementary, I know you used a number of different approaches like linear mixed models or cox-hazard models depending on the data. I understand that you are limited in space and cannot present many details about all the analyses in the main MS. However, just adding the name of the test in front of your statistical estimates presented in the results text would give readers some idea about analysis. They can then consult the supplementary for more details if needed.

L 258 and following: please be aware that you used Gaussian instead of a Poisson distribution to analyse number of incorrect choices across sessions. This should be corrected.

L 325-327: I wonder if you need to add a description of the training to the methods description of the cylinder task. I am familiar with the task and know how it works but other readers might not know that there is a training phase with an opaque cylinder. This is also an important part of the test because this training ensures that what is measured in the transparent phase is actually not confounded with learning the motor response of detouring. If you have space then I would shortly describe the training as well.

L 329-336: I find this very interesting because, as far as I am aware, nobody really finds a correlation of inhibitory control across tasks. Also, I find it interesting that you have the lowest inter-observer reliability in the cylinder task (as per table S2). I have found similar low inter-observer reliability in this task because it is quite hard to score.

L 363-365 & 369: First, you describe your results in regards to which species performed better in the tasks and then you give a summary about the co-evolution of the cognitive abilities and ecological factors focusing on the hypothesis that generalists having better innovation potential and spatial flexibility. This is a little confusing because above you state that GML, the generalist of the two species, has better spatial learning abilities but do not mention flexibility. Also, stating that MBML learn associative reward contingencies faster misses the fact that this result comes from the visual task. I think this first paragraph could be revised to better describe the results. You have a lot of them and this makes it hard to write a concise but detailed summary. I have found that drawing a mental map can help with this issue.

L 372-376: Maybe exploration of novel space, which is what you measured, does not correlate with home range size. If activity is measured properly, in familiar space, the picture might look quite different.

L277-378: You do not introduce these hypothesis in the main MS but have them quite detailed in the supplementary (predictions). I suggest adding a few sentences explaining them and how your data do not support them.

L 379-392: This is quite the nice paragraph. Very nicely written.

L 393: Again, are you taking about the visual tasks specifically? Because the results from the spatial tasks don't really show this.

L 395-397: Are you referring to the visual reversal learning now? Also, how did you tests that motor control and lower persistence actually contributed to their good performance in the visual learning tasks?

L 450: I would rephrase. "elucidate" is a quite strong word for a study with a sample size of 2.

L 586: Again, what does "in relation" mean?

Figures in general: How did you decide which data to present visually and which not? I would think that this is related to the significance of the result. Why is, for example, the spatial learning not shown in which the GML was faster. Also, you could plot the reversal performance after the discrimination performance in both visual and spatial learning tasks.

Figure 1: I really like it. Very clear!

Supplementary material

The supplementary material includes many details about the testing procedures and analysis. I have therefore some questions and comments I list below:

L58-59: Could you give the age range for each species separately?

L 92: Why did you choose to only double code 10% of videos when the field standard is 20%?

L 133-146: Open field test. You describe the repeated testing of certain individuals to calculate repeatability. I miss the information on how much time (on average) passed between these

repeated measures. You do state further above that lemurs were recaptured within at least 3 nights after their last release. I am just wondering because the time between measurements can greatly affect repeatability measures with short intervals often giving higher repeatability. Did you also retest individuals across years?

L 159-160: Please explain what rears and dips are.

L 161-164: What was used as the response variable “activity” in these models?

L 178-183: Again you do not mention the response variables here. I assume you ran 3 models one for each measure you describe in the paragraph above? Or did you run 4 models also looking at the approach speed? If you ran multiple test to answer the same question about neophilia, did you consider to adjust your alpha levels? I see you do that multiple times but you mostly report the results from the intercept model? Do I interpret your results tables correctly?

L 204-206: I understand that your low samples size in MBML prevented you from analysing your data using complex models. This also means, however, that you again used multiple test to basically answer the same questions. Did you consider adjusting your alpha level?

L 220-223: Are you not worried that giving 17 individuals extra training might have influenced their performance in the following persistence test?

L 233: Could you please repeat how you calculated persistence here? It is not clear from the main manuscript.

L 292: There is an error with the reference

L 298. I think this calculation should at least be mentioned in the main MS

L 306-307: Number of incorrect trials usually follows a Poisson distribution not a Gaussian distribution because it is count data. Also, associative learning ability is not mentioned in the main MS but instead you describe visual discrimination, reversal, spatial discrimination and spatial reversal learning ability. Is this equivalent?

L 356: Do I understand correctly, that you ran 15 trials per night (also in the visual and spatial discrimination and reversal learning task)? Meaning within 3 days you ran 30 trials? This is, unfortunately, not described and missing from the methods description.

L 413-414: Were these holes also present in the opaque cylinder? Generally, what experience do your animals have with transparency, as this can greatly affect performance in this task (amongst a number of other non-cognitive factors which lead to a lot of criticism of the cylinder task).

L 502: Are you sure you mean exclude here? I think you mean include, because by excluding it your sample size increases which you state in the next sentence.

L 579-581: Visual and spatial discrimination and reversal learning. I see that your sample size decreases. I am not sure if you stated this properly above but does that mean that some individuals did not reach the learning criterion or that they could not be recaptured. You might want to give some additional explanation in the table caption about this fact.

L 587-588: I think there is a repeat in the caption: between the first, second, and third test, and between the first, second, and third test.

Results Tables: I am wondering, based on how you describe it I assumed that you only presented the results for fixed effects if the comparison with the null model was significant, so they were different. But in table S5, neophilia, it is above 0.05 (same for table S13, S15). In table S18 and S19 results of fixed effects are only given if the comparison was significant. Why did you choose to not do this in the inter-species models? I might have not fully understood your selection process about what to report. I think this could be improved.

L 675: I think your arrow symbols got lost in the table caption. I see an \bar{n} and a δ in the caption but arrows in the table.

Data file

The datafile is a summary table not a raw datafile. If I understood your analyses correctly, for some models you used trial by trial data or at least session by session data which is not present in the provided datafile, or at least I think so. A readme file which explains your columns would also be great as the column names are not all self-explanatory. Finally, you missed the opportunity to provide r code that can be used to reproduce your results. Therefore, unfortunately, the reproducibility of your study is diminished.

Overall, I think this is a great study and well written manuscript. This will be of great interest to a general audience because it reveals quite interesting results in regards to the ecology of the species. I applaud the dedication of the researcher to conduct such a long and complex study with wild caught animals!

Birgit Szabo

Author's Response to Decision Letter for (RSPB-2021-1728.R0)

See Appendix A.

Decision letter (RSPB-2021-1728.R1)

01-Nov-2021

Dear Dr Henke-von der Malsburg

I am pleased to inform you that your manuscript entitled "Linking cognition to ecology in wild sympatric mouse lemur species" has been accepted for publication in Proceedings B.

Data Accessibility section

Open Access

Paper charges

Sincerely,
Dr Robert Barton
Editor, Proceedings B
mailto: proceedingsb@royalsociety.org

Associate Editor:

Comments to Author:

Thank you for the careful revision. I appreciate the thorough and well-organized response to the previous reviews. I am happy to now accept this manuscript for publication. Thank you for submitting to Proceedings B.

Appendix A

Deutsches Primatenzentrum GmbH ■ Leibniz-Institut für Primatenforschung
Kellnerweg 4 ■ 37077 Göttingen ■ Germany

Proceedings of the Royal Society B

Contact: Johanna Henke-von der Malsburg
Department: Behavioral Ecology and Sociobiology
Telephone: +49 551 3851-471
E-mail: JMalsburg@dpz.eu
Web: www.dpz.eu

Date: 17.11.2021
Place: Göttingen

Ref.: MS ID RSPB-2021-1728

Evolution of primate cognition: linking cognition to ecology in wild sympatric mouse lemur species

Dear Professor Barton,

please find enclosed our revised manuscript entitled 'Linking cognition to ecology in wild sympatric mouse lemur species' that we would like to re-submit for publication in *Proceedings of the Royal Society B*.

We modified our manuscript in response to the constructive comments and suggestions of the associate editor and the two reviewers. Below, we answer them in the order of their appearance in the paper (Reviewers' comments in green, our comments in black, modified text passages in grey). Additionally, we fixed some minor spelling errors and references for the tables in the ESM and complemented our supplementary data files.

General comments

❖ **Associate editor:** We have obtained two reviews of your manuscript, and I am happy to tell you that both the reviewers liked your paper. My own reading of your manuscript supports the reviews: this is an interesting piece of work that could become publishable in *Proceedings B*. However, although the reviews were generally very positive, one of the reviewers came up with a long list of useful suggestions and clarifying questions, which should lead to an improved manuscript. The comments by the other reviewer are fewer and perhaps easier to deal with, but they too would likely improve your paper. I am of the same impression as the reviewers that you could improve the manuscript with several changes including providing more methodological detail, justifying and explaining your statistical choices more clearly, and interpreting the somewhat conflicting results more carefully. Review 2 is correct to note that it can be difficult to summarize so many experiments both in detail and concisely but the reviewer makes some suggestions on what needs clarifying. I look forward to seeing your revised manuscript at which time I will decide if it needs to be seen by the reviewers again.

Thank you for this overall positive assessment. We will address all the points highlighted here in our specific responses to the reviewers' comments below.

Bank Details
Sparkasse Göttingen, BIC: NOLADE21 GOE
IBAN: DE16 2605 0001 0018 0020 22

Office of Company
Göttingen

Management
Prof. Dr. Stefan Treue
Dr. Katharina Peters

Chairman of Supervisory Board
MDirig. Rüdiger Eichel

Commercial register
Göttingen HRB 933

Tax No.
StNr.: 20/206/03084
UStIdNr.: DE115314015

- ❖ **Reviewer 1:** This manuscript evaluates species differences on a battery of cognitive tasks implemented with two sympatric mouse lemur species. The main question is whether variation in ecological features (generalist versus specialist) explains differences in cognitive abilities between these species. Results showed that the more generalist species exhibited better spatial learning and innovated more than the more specialist species. In general, I found this study to be well-designed, and analyzed appropriately, with results that can contribute to our understanding of cognitive evolution. However, I have few comments for the authors' consideration.
- ❖ **Reviewer 2:** This manuscript describes a study looking at cognition and personality across two mouse lemur species. The study spans multiple field seasons, high samples sizes (at least in one species) and uses a large number of cognitive tests supplemented by a few personality tests. I want to commend the authors for this great study. It is quite extensive and will be a great addition to the literature. I found the manuscript well written and therefore easy to follow. I have a few comments and questions regarding the main manuscript and the supplementary which details most of the testing procedures and statistical analyses.
- ❖ **Reviewer 2:** Overall, I think this is a great study and well written manuscript. This will be of great interest to a general audience because it reveals quite interesting results in regards to the ecology of the species. I applaud the dedication of the researcher to conduct such a long and complex study with wild caught animals!

We are happy about this overall very positive feedback by the two reviewers. Further, we appreciate the helpful comments and suggestions pointing out where our wording was confusing or needed further explanation. We now detail our statistical report in the main text, provide more methodological detail in the ESM, and present the (insignificant) results in the supplementary tables that we have not shown before. Additionally, we carefully corrected the summary of our results and the following discussion. The specific changes are listed below in response to the specific comments.

Title:

- ❖ **R#2:** I think the title is catchy but also an overstatement. "Evolution of primate cognition" would fit better as the title for a big comparative study looking at many primate species. In your study you only looked at 2. This means your sample size in regards to your statement on the evolution of primate cognition is 2 and insufficient. I suggest rephrasing. The rest of the title is great!

We followed the suggestion and changed the title to "Linking cognition to ecology in wild sympatric mouse lemur species".

Main manuscript:

- ❖ **R#2:** L 35-42: This is a nice introductory paragraph. I am just a little worried about the generality of the statements while mostly primate literature is cited. I suggest either making the statements more primate centered or adding literature from other taxonomic groups.

We totally understand these concerns. To date, relevant research has focused mainly on primates and birds, as reviewed in Henke-von der Malsburg et al. (2020) *Behav Ecol Sociobiol*. However, we followed your suggestion and adjusted our wording in LL37-42: "Recent comparative analyses across primates suggested that evolutionary variation in brain size is

better predicted by ecological than social factors(3). Yet, little is known about whether and how these factors are linked to performance in cognitive tests in primates, but also across other taxonomic groups(4,5). Hence, to better understand the evolution of cognitive abilities and the underlying variation in brain size, studies of how variation in specific ecological or social factors are linked to performance in cognitive tests across taxa are required.”

- ❖ **R#2:** L 81: You talk about 1,104 experiments here but use tests instead of experiments across the rest of the manuscript (and the supplementary material). I suggest using tests.

We followed this suggestion and changed “experiments” in L81 and in the abstract to “tests”.

- ❖ **R#2:** L 82: What you measured with your open field test is actually exploration because activity can only be measured in a familiar environment while exploration occurs in unfamiliar space. I suggest you change this in the main MS and the supplementary material.

We chose to label the behavior we measured in the open field test as “activity” since we measured the duration the animals spent locomoting in the arena. However, we understand this important note of reviewer #2 and changed “activity” throughout the MS and the ESM to the standard wording “exploration”.

- ❖ **R#1:** The introduction on Line 83-86 reports the logic behind the choice of the cognitive tasks administered. It is not immediately clear what the authors mean when they say that inhibitory control, means-end understanding, and goal-directedness are not obvious ecological relevant. Did the authors mean that these abilities can also be used in contexts involving social rather than feeding decisions? A similar argument could be made for the other kind of tasks since associative and flexible learning or spatial memory could also be used in social situations. Perhaps this could be clarified.

With “not ecologically relevant”, we mean that variation in these cognitive abilities are not expected in relation to the degree of ecological specialization. For example, both study species do not rely on specific extractive foraging techniques, which are usually related to these cognitive abilities. A respective clarification is now included in the main text LL85-86 (“i.e., cognitive abilities that are not expected to covary with the degree of ecological specialization”), LL440-441 (“In tasks with little ecological relevance, where species differences in cognitive performance were not predicted regarding the degree of ecological specialization”, and in the supplementary text LL37-38 (“[...] in which differences in performance are not predicted by the degree of ecological specialization”).

- ❖ **R#1:** From the method and the Supplementary materials is not clear whether each subject received all the tasks of the battery or different subjects were tested in different tasks (between-subjects procedure). Please clarify.

We are sorry, that this information was not clear in the original description of the methods. In LL80-82, we stated “Using a comprehensive test battery with ten cognitive tests and two standard personality tests, we compared cognitive abilities of these two species. In a total of 1,104 tests, we tested N=120 GML and N=34 MBML.” Table S1 shows the different sample sizes across the tasks. We added a clarification to the MS in LL112-114 and the caption of Table S1 in the ESM: “Sample sizes differ between tasks as it was not possible to recapture all individuals until they have participated in all tasks of the test battery (ESM, Table S1).” We hope that it becomes now clearer that we tested the same individuals in different tasks, but that not all individuals performed each task.

- ❖ **R#2:** L 112: Do you have any evidence that your two lemur species cannot perceive red light in the wavelength your light is emitting?

A recent study including samples of both mouse lemur species from our study site concluded: “Thus, we assume that these *Microcebus* species are color-blind in the central part of their visual field and have dichromatic peripheral vision. The prediction is that they cannot use color vision for the detection of food items, which supposedly requires binocular focusing with the central area. It is unclear what advantage their peripheral color vision offers.” (Peichl et al. (2019) *J Comp Neurol*). We included a respective statement in the ESM, LL65-66 (“under red light conditions, which wavelengths is not visible for the dichromatic mouse lemurs (11”).

❖ **R#2:** L 131: what was “the response” exactly?

We added an explanation in L134: “the response (i.e., entering the experimental platform and visualizing the task)”.

❖ **R#2:** L 139: what does “in relation” mean here exactly? Could you make this description more clear?

We extended the explanation of persistence in LL141-142: “We calculated an individual’s persistence rate by dividing the duration manipulating the box by the duration being in contact with the box”

❖ **R#2:** L 142-157: what was the inter-trial interval, the inter-session interval and how did you calculate the transfer index.

We added an explanation for the intervals in the ESM LL277-285 (“The time-interval between trials within a session was 10-30 s, depending on how fast the experimenter could re-bait the tube, change the tube positions and clean the apparatus, as well as how easily the subject was lured back to the starting position. We defined a trial as correct, if the subject extracted the food reward by only manipulating the S+-form. If it manipulated an incorrect form, we noted the trial as incorrect and let the subject continue to explore the apparatus until it retrieved the reward or a maximum of 10 min. Depending on the subject’s motivation (i.e., actively moving around and easily lured to the start position), we conducted two sessions in a row with a short break of 5 min between sessions and a break of at least 30 min to a third session. We conducted a total of four to five sessions per individual and night, always taking the subject’s motivation into account.”) and the equation for the transfer index in the MS L162: “Equation 1

$$TI = \frac{\text{post-reversal performance}}{\text{pre-reversal performance}}$$

❖ **R#2:** L 166: I am not sure you can call the cylinder task “classical”. Just state that you followed MacLean et al.

We followed this suggestion and deleted “classical” (L171).

❖ **R#2:** L 325-327: I wonder if you need to add a description of the training to the methods description of the cylinder task. I am familiar with the task and know how it works but other readers might not know that there is a training phase with an opaque cylinder. This is also an important part of the test because this training ensures that what is measured in the transparent phase is actually not confounded with learning the motor response of detouring. If you have space then I would shortly describe the training as well.

We are aware of the importance of the training in this task and followed the suggestion by adding a clarification in the MS in LL171-172 “After an initial training session with an opaque cylinder (see ESM), we conducted the experimental session using a transparent cylinder.” The exact training procedure is explained in more detail in the ESM LL415-429.

❖ **R#2:** L 198: With null model you mean an intercept only model? I think you could give this information here. I would also really appreciate if you could add a justification why you did compare full and null models and when you report results from the full models.

We followed the suggestion and extended the respective sentence in LL207-209: “To test the significance of the predictors as a whole, we compared all full models with the respective null model comprising only the intercept and potential random factors (see ESM; (33)).”

- ❖ **R#1:** To calculate the interspecific G-factor, the authors implemented principal axis factor analyses, while to calculate the intra-specific g-factor, they implemented PCAs. What is the logic behind this choice?

It has been suggested that for interspecific comparisons a PAF for the G-factor analysis provides more meaningful results than a PCA (see Budaev 2010 *Ethology*). However, a PCA is usually used to analyse intraspecific g-factors, (e.g., Herrmann & Call 2012 *Phil Trans R Soc B*, Isden et al. 2013 *Anim Behav*, Shaw et al. 2015 *Anim Behav*; but see Shaw & Schmelz 2017 *Anim Cogn*). Therefore, we used both approaches to be comparable to other studies.

- ❖ **R#2:** Results generally: I think it would be a good idea to add the type of test used to generate your results in the main manuscript. After reading the supplementary, I know you used a number of different approaches like linear mixed models or cox-hazard models depending on the data. I understand that you are limited in space and cannot present many details about all the analyses in the main MS. However, just adding the name of the test in front of your statistical estimates presented in the results text would give readers some idea about analysis. They can then consult the supplementary for more details if needed.

We added the statistical tests to the methods in LL193-196 of the MS and in LL96-100 the ESM (“gaussian linear models (LM), gaussian linear mixed models (LMM), negative binomial models (NBM), negative binomial mixed models (NBMM), zero-inflated negative binomial models (0-infl NBM), cox-proportional hazards models (cox PHM), poisson models (PM)”) and the respective abbreviations to the results (e.g. in L233: “LM: $p=0.018$ ”).

- ❖ **R#2:** L 258 and following: please be aware that you used Gaussian instead of a Poisson distribution to analyse number of incorrect choices across sessions. This should be corrected.

We appreciate the statistical advice and corrected our analyses. To investigate variation in visual and spatial discrimination and reversal learning, we calculated negative binomial mixed models (poisson models appeared to be overdispersed) and corrected the corresponding statistics in the main text (see LL260-313), as well as in the supplementary methods (LL327-329) and results (Tables S8-S11).

- ❖ **R#2:** L 329-336: I find this very interesting because, as far as I am aware, nobody really finds a correlation of inhibitory control across tasks. Also, I find it interesting that you have the lowest inter-observer reliability in the cylinder task (as per table S2). I have found similar low inter-observer reliability in this task because it is quite hard to score.

We thank reviewer #2 for this comment. We were also puzzled by the low inter-observer reliability and the missing correlations which suggests, that the cylinder task still offers room for improvement.

- ❖ **R#2:** L 363-365 & 369: First, you describe your results in regards to which species performed better in the tasks and then you give a summary about the co-evolution of the cognitive abilities and ecological factors focusing on the hypothesis that generalists having better innovation potential and spatial flexibility. This is a little confusing because above you state that GML, the generalist of the two species, has better spatial learning abilities but do not mention flexibility. Also, stating that MBML learn associative reward contingencies faster misses the fact that this result comes from the visual task. I think this first paragraph could be revised to better describe the results. You have a lot of them and this makes it hard to write a concise but detailed summary. I have found that drawing a mental map can help with this issue.

We are sorry to hear that it was difficult to follow the summary of our results. We rephrased it by being more specific about the individual cognitive abilities in our discussion and hope that it is now easier to comprehend (e.g., LL375-386: “Gray and Madame Berthe’s mouse lemurs differed in 13 of 21 performance measures (Table 1), which were moderately repeatable in the majority of tests and, hence, consistent within individuals (ESM, Table S2). Overall, the ecological generalist GML were more innovative, more persistent, showed better spatial learning and memory, and greater flexibility when spatial stimulus-reward contingencies were changed. The ecologically more specialized MBML were more explorative and learned visual and spatial reward contingencies faster, achieving better total performances in visual association learning. However, the two species did not differ in neophilia, inhibitory control, means-end understanding or goal directedness. Moreover, we did not find evidence for an interspecific G- or intraspecific g-factor. In summary, our study provides support for domain-specific cognitive co-evolution with ecological factors, and a particular advantage of generalists in confronting novel challenges with a greater innovative potential and greater flexibility when confronted with changing spatial stimuli.”, LL395-396: “GML were indeed more innovative and achieved higher flexibility scores, at least under changing spatial stimuli.”, LL469-473: “The ecologically more generalist species was particularly more innovative, persistent, exhibited better spatial learning abilities, spatial memory, as well as spatial flexibility than the specialist, affording them with the behavioral flexibility to respond adaptively to rapidly changing habitats and anthropogenic disturbances.”).

Additionally, we included the summary table (previously: ESM, Table S20) now in the main manuscript: Table 1.

- ❖ **R#2:** L 372-376: Maybe exploration of novel space, which is what you measured, does not correlate with home range size. If activity is measured properly, in familiar space, the picture might look quite different.

This notion appears plausible. Since we measured exploration instead of activity, we deleted the correlation with home range size (L391).

- ❖ **R#2:** L377-378: You do not introduce these hypothesis in the main MS but have them quite detailed in the supplementary (predictions). I suggest adding a few sentences explaining them and how your data to not support them.

We now explain the hypotheses in more detail in LL391-394: “In addition, the two species did not differ in their neophilic response, supporting neither the *Dangerous-Niche Hypothesis* (i.e., generalists should be more neophobic as they may encounter more dangerous situations) nor the *Neophobia Threshold Hypothesis* (i.e., generalists should be less neophobic as they may have more diverse prior experiences; 34,35).”

- ❖ **R#2:** L 379-392: This is quite the nice paragraph. Very nicely written.

We thank reviewer #2 for this positive feedback.

- ❖ **R#1:** Lastly, I found some results and, consequently, some parts of the discussion unclear. In particular, from the results section and Table S20 seems that the grey mouse lemurs perform better than the Mme. Berthe’s mouse lemurs only in the tasks measuring innovation, persistence, and spatial memory, but not in those measuring visual and spatial discrimination and reversal. Thus, it appears that, although less innovative, MBML are quite flexible. Lines 393-397 report that MBML’s relatively low persistence contributes to their higher associative learning abilities. Yet, they performed better than the other species also when the contingencies were switched. Thus, this result does not fully support the hypothesis that generalists species might have greater flexible learning abilities than more specialist species. Some discussion of this pattern of results will help clarify the paper’s main conclusion.

Reviewer #1 summarized the interspecific differences correctly. However, given that MBML were less innovative (a key measure of flexibility) and achieved lower transfer indices when spatial stimuli changed (another measure of flexibility), we conclude that they show relatively lower flexibility compared to GML. It is also correct that MBML learned visual and spatial associative reward contingencies faster, achieving higher overall performances in visual set-ups. However, their initial response to a changing stimulus, estimated by calculating the transfer index, was similar (changing visual stimuli, change between visual and spatial stimuli) or lower (changing spatial stimuli) compared to GML. Thus, overall, GML show greater flexibility than MBML, while MBML seem to be better in implementing more efficient solutions once a problem has changed. This in turn, provides them with advantages under stable environmental conditions and may be related to greater self control, as well as lower persistence to not use previously successful techniques, which relates to our previous study results in Henke-von der Malsburg & Fichtel (2018) *RSOS*. We explained these relations in LL409-413 (“MBML learned the associative reward contingencies faster; a characteristic that allows specialists to experience advantages over generalists in stable environmental conditions(39). Specifically, MBML’s better motor control to abandon previously successful behaviors(40) and their lower persistence to produce a behavior that does not lead to success (this study), may have contributed to their superior performance, at least in visual associative learning experiments.”)

❖ **R#2:** L 393: Again, are you talking about the visual tasks specifically? Because the results from the spatial tasks don’t really show this.

In this case, we first repeated the result that MBML learned all four associative reward contingencies, i.e., both visual and spatial tasks, faster than GML. In the following, we focus on the overall visual associative performance, which we now specified in L412 (“[...] may have contributed to their superior performance, at least in visual associative learning experiments”). Unfortunately, our sample size in the spatial learning tasks are very low for MBML, which may prevent the possibility to draw firm conclusions about potential interspecific differences.

❖ **R#2:** L 395-397: Are you referring to the visual reversal learning now? Also, how did you tests that motor control and lower persistence actually contributed to their good performance in the visual learning tasks?

In this case, we are referring to MBML’s better motor control, as reported in Henke-von der Malsburg & Fichtel (2018) *RSOS*, which we cite at this point, and their better persistence as reported in the present MS. Given our data and sample sizes, we cannot clearly justify how these skills contribute to better visual associative learning abilities, but we suggest a functional relationship, now adding the conditional “may have contributed” (L412).

❖ **R#2:** L 450: I would rephrase. “elucidate” is a quite strong word for a study with a sample size of 2.

We followed this suggestion and changed “elucidate” to “can help to unfold” (see L468).

❖ **R#2:** Figures in general: How did you decide which data to present visually and which not? I would think that this is related to the significance of the result. Why is, for example, the spatial learning not shown in which the GML was faster. Also, you could plot the reversal performance after the discrimination performance in both visual and spatial learning tasks.

In fact, it was not an easy decision to exclude some of the visual summaries of our results. However, as we were limited in space, we decided to focus on the interspecific differences. With regard to spatial learning, we chose to show overall spatial memory performance (now Figure 2G) rather than learning across trials. With regard to the discrimination and reversal learning paradigm, we preferred to show only the learning in the visual discrimination (now Figure 2D), since the figure looks basically the same for the visual reversal and the two spatial parts but contains more data due to the larger sample size in the visual discrimination. Still, we

followed your suggestion and include now the overall performances in all of the four test parts (Figure 2E), as well as the transfer indices (Figure 2F).

❖ **R#2: L 586: Again, what does “in relation” mean?**

We changed the wording “in relation” to “as a rate” (Figure 2C caption) and hope that it is now easier to understand.

❖ **R#2: Figure 1: I really like it. Very clear!**

We are happy that Figure 1 helps the reader understanding the experimental test battery.

Supplementary material

R#2: The supplementary material includes many details about the testing procedures and analysis. I have therefore some questions and comments I list below:

❖ **R#2: L58-59: Could you give the age range for each species separately?**

We followed this suggestion in LL56-57: “The individuals’ ages ranged from 0.26 to 8.58 yrs. (mean=1.21 ± 1.40) in GML and from 0.33 to 5.35 yrs. (mean=1.20 ± 1.17) in MBML.”

❖ **R#2: L 92: Why did you choose to only double code 10% of videos when the field standard is 20%?**

We recorded a total of 2586 videos. Due to time and financial constraints we, unfortunately, did not manage to recruit volunteers to achieve 20% of double-coded videos for each task. To be consistent, we consequently decided to calculate the interobserver reliability of only 10% of double-coded videos, which we chose randomly out of the total number of double-coded videos.

❖ **R#2: L 133-146: Open field test. You describe the repeated testing of certain individuals to calculate repeatability. I miss the information on how much time (on average) passed between these repeated measures. You do state further above that lemurs were recaptured within at least 3 nights after their last release. I am just wondering because the time between measurements can greatly affect repeatability measures with short intervals often giving higher repeatability. Did you also retest individuals across years?**

We agree with this concern about the inter-test time interval and added this information in LL142-146: “The first repetition was usually conducted when subjects were captured for the second time (number of days between tests: mean=81.38 ± 88.02). Further repetitions were conducted each time when we re-captured the subject until it had accomplished the complete test battery, but for a maximum of two times per field season (days between test 2 and 3: mean=130.14 ± 122.12; days between test 3 and 4: mean=66.14 ± 76.32).”

In this paragraph, we already included the information that we tested the individuals a maximum of two times per field season (i.e., a time period of three months) but repeated the testing until the individuals finished the complete test battery. As explained in LL78-79, most individuals have been recaptured two to five times, which could be within one or several field seasons. Most of the individuals have been tested in one year only; first, because we focused on finalizing the test battery of a subject towards the end of one year. Second, because mouse lemurs have a high mortality rate which did not allow us to recapture all individuals across field seasons or years (see also Table S1 and S3 for sample sizes).

Finally, we report results only for test 1 to 4, since the sample sizes in further tests decreased due to the above reasons. We added this information now in LL148-149: “Only a small subset was tested in a fourth (N=29), fifth (N=8), sixth (N=4) or seventh (N=1) open field test.”

❖ **R#2:** L 159-160: Please explain what rears and dips are.

We followed this suggestion in LL164-164’5: “with rears, i.e., standing on hindlegs, (log-transformed) and dips, i.e., inserting head into blind holes or closed doors of the open field”

❖ **R#2:** L 161-164: What was used as the response variable “activity” in these models?

We added this information in LL167-168: “To estimate variation in exploration, we used a linear model with gaussian error distribution (‘lm’ function) with the duration subjects spent locomoting as response variable.”

❖ **R#2:** L 178-183: Again you do not mention the response variables here. I assume you ran 3 models one for each measure you describe in the paragraph above? Or did you run 4 models also looking at the approach speed? If you ran multiple test to answer the same question about neophilia, did you consider to adjust your alpha levels? I see you do that multiple times but you mostly report the results from the intercept model? Do I interpret your results tables correctly?

We ran only 1 model with the contact frequency as response variable (i.e., we did not adjust alpha levels). However, we also measured the approach latency, the contact latency, as well as the approach speed, because latencies have often been used in previous studies as measure for neophilia/neophobia. For terms of completeness, we describe these variables as well.

We added the information about the response variable in L188: “[...] the number of contacts with the novel object as response variable”, as well as a clarification why we chose this response in LL184f: “We chose the contact frequency as our measure for *neophilia*, as the other variables were highly skewed towards individuals that did not contact the object.”

The result tables show 1) the results of the intercept model (first line, bold), and 2) the results of the reduced models for each factor separately.

❖ **R#2:** L 204-206: I understand that your low samples size in MBML prevented you from analysing your data using complex models. This also means, however, that you again used multiple test to basically answer the same questions. Did you consider adjusting your alpha level?

We understand this concern. Since corrections for multiple testing are also debated because the likelihood of type II errors also increases, so that truly important differences are deemed non-significant, we followed the recommendation by Perneger (1998) *BMJ* and only report the results with corrections for multiple testings.

❖ **R#2:** L 220-223: Are you not worried that giving 17 individuals extra training might have influenced their performance in the following persistence test?

Indeed, different training procedures can influence subsequent test performances and we carefully controlled for irregularities. In the case of the persistence test, the training should not serve to learn a motor response (i.e., opening the well) but to signal that the animals can extract a food reward from the box (i.e., to motivate the animals to manipulate the box). Consequently, we did not expect a difference in persistence. However, we now included the results of the respective model excluding these individuals in Table S7 (model: b) and LL248-250: “Running the same model excluding the N=17 individuals that received additional training, did not affect the results (Table S7, model: b).”

- ❖ **R#2:** L 233: Could you please repeat how you calculated persistence here? It is not clear from the main manuscript.

We are sorry that our wording was not clear. In the preceding paragraph, we now provide a description: “We calculated an individual’s persistence as the duration manipulating the box in relation to the duration being in contact with the box” (LL235ff), i.e., persistence is the duration manipulating divided by duration in contact.

- ❖ **R#2:** L 292: There is an error with the reference

We fixed the reference in L309.

- ❖ **R#2:** L 298. I think this calculation should at least be mentioned in the main MS

We followed this suggestion in the MS L162 (see above).

- ❖ **R#2:** L 306-307: Number of incorrect trials usually follows a Poisson distribution not a Gaussian distribution because it is count data. Also, associative learning ability is not mentioned in the main MS but instead you describe visual discrimination, reversal, spatial discrimination and spatial reversal learning ability. Is this equivalent?

As mentioned above, to investigate variation in visual and spatial discrimination and reversal learning, we ran now negative binomial mixed models (poisson models appeared to be overdispersed) and corrected respective statistics in the main text (see LL260-313), as well as in the methods (LL327-329) and results (Tables S8-S11) in the ESM.

When we were referring to associative learning abilities more generally, we named it accordingly (e.g. in L83, LL145ff, L377, LL405ff). Otherwise, we were more specific, naming the exact task (e.g. L375, L424).

- ❖ **R#2:** L 356: Do I understand correctly, that you ran 15 trials per night (also in the visual and spatial discrimination and reversal learning task)? Meaning within 3 days you ran 30 trials? This is, unfortunately, not described and missing from the methods description.

This comment concerns the plus maze, which covers one session of 15 trials which were usually ran in a row, i.e., in one night. We extended our description in L380: “15 successive trials”

For the discrimination and reversal learning tasks, we now added inter-trial and inter-session intervals as suggested and described above.

- ❖ **R#2:** L 413-414: Were these holes also present in the opaque cylinder? Generally, what experience do your animals have with transparency, as this can greatly affect performance in this task (amongst a number of other non-cognitive factors which lead to a lot of criticism of the cylinder task).

For the training session, the opaque cylinder did not have holes in its front through which the animals could smell the food reward. We chose this modification to the transparent test cylinder, as the training served only to learn the motor response, i.e., to make the detour to one of the sides of the cylinder.

Our subjects did, in general, not have any experience with transparency, except for N=2 GML that were not naïve to the task (see Table S1).

- ❖ **R#2:** L 502: Are you sure you mean exclude here? I think you mean include, because by excluding it your sample size increases which you state in the next sentence.

We are sorry that this section was difficult to read. “exclude” is correct – similar to the PAF, we excluded the performance data of the spatial discrimination and spatial reversal for the first

PCA to obtain a larger sample size (N=23 GML, N=9 MBML) compared to the inclusion of the two tasks (N=21 GML, N=4 MBML). We hope that these changes improved our wording in LL519-521: “The first PAF contained the performance scores of individuals that completed all tests (excluding the spatial discrimination and the spatial reversal due to very low sample sizes in these tasks; N=21 GML, N=9 MBML).”, as well as in LL526-529: “The first PCA per species contained the performance scores of all tests (excluding the spatial discrimination and the spatial reversal due to very low sample sizes in these tasks), which resulted in a data set of N=23 GML and N=9 MBML.”

❖ **R#2:** L 579-581: Visual and spatial discrimination and reversal learning. I see that your sample size decreases. I am not sure if you stated this properly above but does that mean that some individuals did not reach the learning criterion or that they could not be recaptured. You might want to give some additional explanation in the table caption about this fact.

This is an important point. However, we explained the differences in sample sizes already above, see LL320-324 (“The sample sizes slightly decreased over the experimental tasks (Table S1), since we failed to recapture some individuals before they completed the repeated discrimination and reversal learning paradigm. Only N=1 individual failed to learn the reward contingency in the spatial discrimination. For N=2 individuals, we erroneously continued with the subsequent experimental task before the previous reward contingency was learned, and they were subsequently excluded from the analyses.”). Now, we added an additional explanation in the caption of Table S1 following an earlier comment: “Sample sizes differ between tasks as it was not possible to recapture all individuals until they had completed all tasks of the test battery”

❖ **R#2:** L 587-588: I think there is a repeat in the caption: between the first, second, and third test, and between the first, second, and third test.

We want to thank reviewer #2 for this observation and corrected the caption of Table S3 to “Intraclass correlation coefficients were calculated between the first and second test, between the first, second, and third test, and between the first, second, third, and fourth test.”

❖ **R#2:** Results Tables: I am wondering, based on how you describe it I assumed that you only presented the results for fixed effects if the comparison with the null model was significant, so they were different. But in table S5, neophilia, it is above 0.05 (same for table S13, S15). In table S18 and S19 results of fixed effects are only given if the comparison was significant. Why did you choose to not do this in the inter-species models? I might have not fully understood your selection process about what to report. I think this could be improved.

We apologize for this inconsistency in our results report. As we focused on the interspecific comparison, we reported all results for these models. To make the huge amount of the intraspecific results better readable, we only included statistics for the individual predictors if the full-null model comparison was significant. We now completed this information (see Table S17, S18). Additionally, we improved the tables mentioning the specific tasks above the response variable, and we fixed erroneous references to these tables.

❖ **R#2:** L 675: I think your arrow symbols got lost in the table caption. I see an ñ and a ò in the caption but arrows in the table.

We want to thank reviewer #2 for this comment and corrected the caption of this Table (formerly Table S20 in the ESM, now Table 1 in the MS) using the arrow symbols.

Data file

❖ **R#2:** The datafile is a summary table not a raw datafile. If I understood your analyses correctly, for some models you used trial by trial data or at least session by session data which is not present in the provided datafile, or at least I think so. A readme file which explains your columns would also be great as the column names are not all self-explanatory. Finally, you missed the opportunity to provide r code that can be used to reproduce your results. Therefore, unfortunately, the reproducibility of your study is diminished.

We thank reviewer #2 for these remarks. We added supplementary data files containing not only the general intelligence data (previous data file), but also the performance and learning data of the individual tests, as well as the data of our interobserver reliability and intraindividual repeatability analyses.

However, the R code is quite comprehensive and might be difficult to follow. Therefore, we prefer to provide only the relevant R code upon specific request..

We hope that you and the referees are satisfied by the revised manuscript, and we look forward to hearing about your decision regarding its publication.

Sincerely,

Johanna Henke-von der Malsburg